# MaskSAM: Auto-prompt SAM with Mask Classification for Medical Image Segmentation

## Abstract

Segment Anything Model (SAM), a prompt-driven foundation model for natural image segmentation, demonstrated impressive zero-shot performance. However, SAM does not work when directly applied to medical image segmentation tasks, since SAM lacks the functionality to predict semantic labels for predicted masks and needs to provide extra prompts, such as points or boxes, to segment target regions. Meanwhile, there is a significant gap between 2D natural images and 3D medical images, so the performance of SAM is imperfect for medical image segmentation tasks. Following the above issues, we propose MaskSAM, a novel mask classification prompt-free SAM adaptation framework for medical image segmentation. We design a prompt generator combined with the image encoder in SAM to generate a set of auxiliary classifier tokens, auxiliary binary masks, and auxiliary bounding boxes. Each pair of auxiliary mask and box prompts, which addresses the requirements of extra prompts, is associated with class label predictions by the sum of the auxiliary classifier token and the learnable global classifier tokens in the mask decoder of SAM to solve the predictions of semantic labels. Meanwhile, we design a 3D depth-convolution adapter for image embeddings and a 3D depth-MLP adapter for prompt embeddings. We inject one of them into each transformer block in the image encoder and mask decoder to enable pre-trained 2D SAM models to extract 3D information and adapt to 3D medical images. Our method achieves state-of-the-art performance on AMOS2022 Ji et al. (2022), 90.52% Dice, which improved by 2.7% compared to nnUNet. Our method surpasses nnUNet by 1.7% on ACDC Bernard et al. (2018) and 1.0% on Synapse Landman et al. (2015) datasets.

## 1 Introduction

Foundation models Devlin et al. (2018); He et al. (2022), trained on vast and diverse datasets, have shown impressive capabilities in various tasks OpenAI (2023b); Radford et al. (2021) and are revolutionizing artificial intelligence. The extraordinary zero-shot and few-shot generalization abilities of foundation models derive a wide range of downstream tasks and achieve numerous and remarkable progress. In contrast to the traditional methods of training task-specific models from scratch, the "pre-training then finetuning" paradigm has proven pivotal, particularly in the realm of computer vision. Segment Anything Model (SAM) Kirillov et al. (2023), pre-trained over 1 billion masks on 11 million natural images, was recently proposed as a visual foundation model for prompt-driven image segmentation and has gained significant attention. SAM can generate precise object binary masks based on its impressive zero-shot capabilities. As a crucial branch of image segmentation, medical image segmentation was dominated by deep learning medical segmentation methods Ronneberger et al. (2015); Akkus et al. (2017); Avendi et al. (2016) for the past few years. The existing deep learning models are often tailored for specific tasks and achieve remarkable progress due to the consumption of a strong inductive bias. This raises an intriguing question: Can SAM still have the ability to revolutionize the field of medical image segmentation? Or can SAM still achieve high-performance results in medical image segmentation by properly fine-tuning based on SAM's strong zero-shot capabilities in natural image segmentation?

Since the publication of SAM, numerous studies have attempted to adapt it for medical image segmentation; however, few SAM-based models have effectively addressed the medical challenges, such as the AMOS22 challenges Ji et al. (2022), a key benchmark for validating medical image

Figure 1: The overview architecture of our proposed MaskSAM.

segmentation models. The limitations of existing SAM-based methods in tackling these challenges can be attributed to three main factors:

Inability to predict semantic labels: SAM generates a single binary mask per prompt without associating semantic labels, which is insufficient for medical images that often contain multiple labels with essential semantic information. Existing SAM-based methods typically produce binary (one-class) segmentation. They can be categorized into two groups: The first group retains the original SAM structure, using it directly or fine-tuning portions of it on target datasets, as seen in MedSAM Ma et al. (2024), Polyp-SAM Li et al. (2024), and SAM.MD Wald et al. (2023). These methods require additional prompts to evaluate performance on medical datasets. The second group modifies SAM but fails to implement semantic labeling, such as DeSAM Gao et al. (2023), AutoSAM Shaharabany et al. (2023), and 3DSAM-Adapter Gong et al. (2023), sacrificing SAM's zero-shot capabilities and restricting their functionality to one-class datasets.

Inadequate handling of prompt requirements: SAM demands precise user input prompts to segment target regions, a requirement not addressed by many models. Typically, these models rely on ground truth (GT) data to generate prompts during inference, as seen in Med-SA Zhang et al. (2024), 3DSAM-Adapter Gong et al. (2023), and SAM-U Deng et al. (2023a). However, these methods cannot participate in challenges where labels for prompts are unavailable.

Subpar performance: Even when using appropriate prompts, SAM often underperforms in medical image segmentation tasks. Although some models, like SAMed Zhang & Liu (2023) and SAM3D Bui et al. (2024), attempt to address semantic label generation and prompt requirements, their performance remains inferior. Consequently, refining the fine-tuning process for adapting SAM from natural image segmentation to medical image segmentation is crucial, contributing to the limited participation of SAM-based models in medical challenges. Additional related work is presented in the Appendix A.

Our proposed SAM-based MaskSAM effectively addresses the aforementioned challenges and achieves state-of-the-art performance in the AMOS22 challenge. To handle the extra prompt requirements, we designed a prompt-free architecture for SAM. We observed that the image encoder utilizes the Vision Transformer (ViT)Dosovitskiy et al. (2020), pre-trained with a masked auto-encoderHe et al. (2022), as its backbone. Leveraging ViT's robust representation capabilities, the image encoder extracts essential features through a series of transformer blocks. To eliminate the dependency on manual prompts, we introduce a prompt generator that employs multiple levels of feature maps from the image encoder, generating a set of auxiliary binary masks and bounding boxes as prompts, thus resolving the need for additional prompts.

To enable semantic label prediction, the prompt generator simultaneously produces a set of auxiliary classifier tokens. Since the mask decoder lacks inherent classifier tokens for outputting class predictions, we were inspired by MaskFormer Cheng et al. (2021) to introduce global learnable classifier tokens. These tokens, combined with the auxiliary classifier tokens, associate each predicted binary mask with its corresponding class in the mask decoder.

We also designed a dataset mapping pipeline, as illustrated in Figure 1, that converts multi-class masks into sets of binary masks with semantic labels. This pipeline accommodates the varying lengths of binary masks in the ground truth data. Drawing inspiration from DETR Carion et al. (2020) and MaskFormer Cheng et al. (2021), our prompt generator creates a sufficient number of

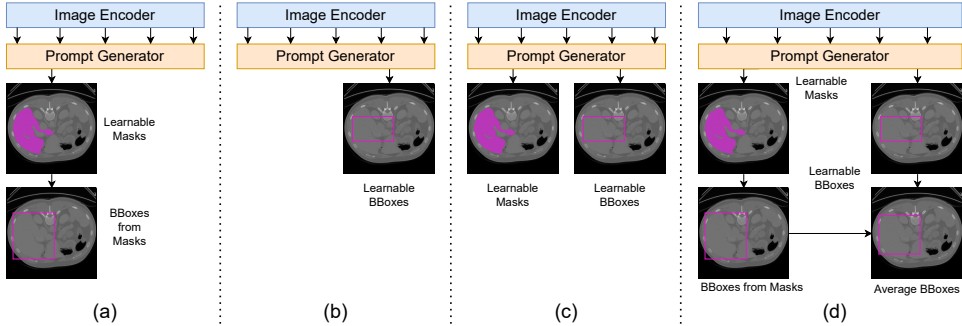

Figure 2: (a) A prompt generator with learnable masks. (b) A prompt generator with learnable boxes. (c) A prompt generator with learnable masks and learnable boxes. (d) A prompt generator with learnable masks and average boxes.

prompts, exceeding the maximum number of class-level binary masks within the dataset. Bipartite matching is then used to align predicted masks with ground truth segments, ensuring accurate loss calculation.

While SAM addresses some functionality issues, it does not consistently perform well in medical image segmentation tasks, even with appropriate prompts. Numerous studies Deng et al. (2023b); Hu & Li (2023); Zhou et al. (2023); Mohapatra et al. (2023); Roy et al. (2023); Wang et al. (2023); He et al. (2023) have shown that SAM struggles, particularly in cases with weak boundaries, low contrast, or small and irregular shapes, as supported by other investigations Ji et al. (2023a;b). Consequently, fine-tuning SAM for medical image segmentation has become a primary focus. However, fine-tuning SAM, a large model, demands substantial computational resources.

Several studies Ma & Wang (2023); Wu et al. (2023); Li et al. (2023); Gong et al. (2023) have demonstrated the effectiveness of efficient fine-tuning by incorporating lightweight adapters Houlsby et al. (2019) for medical image segmentation tasks. In our approach, we employ these lightweight adapters for efficient fine-tuning. However, existing methods often exclude the prompt encoder or mask decoder to circumvent the need for additional prompts, which disrupts SAM's inherent consistency and overlooks the valuable components trained on large-scale datasets.

To address this challenge, we maintain the complete structure of SAM by preserving all components and freezing their weights, while strategically inserting our designed blocks for adaptation. This approach enables us to retain SAM's zero-shot capabilities while effectively adapting it to medical image segmentation tasks. In this paper, we employ the lightweight adapter for efficient fine-tuning. However, the above works usually abandon the prompt encoder or mask decoder to avoid the requirements of additional prompts provided, which would destroy the consistent system of SAM and abandon the robust prompt encoder and mask decoder, which are trained via large-scale datasets and lots of resources. Therefore, the primary challenge lies in modifying the structure to preserve the inherent capabilities of SAM. Therefore, we keep all structures, freeze all weights, and only insert designed blocks into SAM to adapt. In this way, we retain the zero-shot capabilities of SAM and adapt SAM to medical image segmentation.

Unlike classic 2D natural images, many medical scans, such as MRI and CT, are 3D volumes with an additional depth dimension. To incorporate this depth information, we introduce learnable layers designed for the extra depth dimension in the lightweight adapters.

In SAM, both the image encoder and mask decoder consist of transformer blocks where adapters can be inserted. The mask decoder includes two types of attention blocks: one for prompt embeddings and another for image embeddings. The original adapter processes only the last (channel) dimension, which limits its ability to capture relationships among tokens. Since image embeddings contain crucial spatial information, it is es-

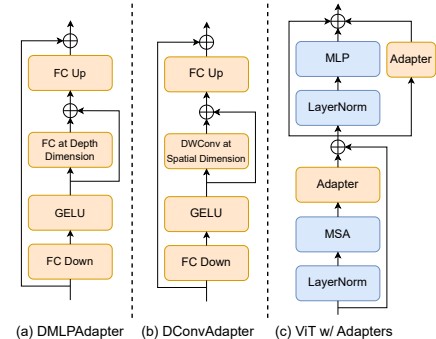

Figure 3: The proposed adapters.

sential for the model to understand spatial relationships. To address this, we developed a 3D depth-convolution adapter (DConvAdapter) that integrates a 3D depth-wise convolution layer with a skip connection within the original adapter, specifically for attention blocks handling image embeddings.

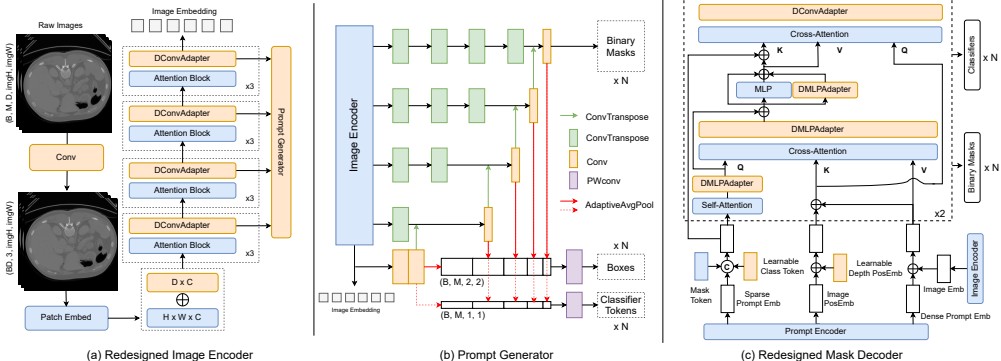

Figure 4: Overview of (a) redesigned image encoder, (b) proposed prompt generator, and (c) redesigned mask decoder. Blue and white boxes are frozen and the rests are tuned.

For the remaining attention blocks involving prompt embeddings, we introduce a 3D depth-MLP adapter (DMLPAdapter), which incorporates an inverted-bottleneck architecture with two fully connected (FC) layers and an activation layer to process the depth dimension, also featuring a skip connection. This enables the adapter to learn additional depth information. These designed adapters are illustrated in Figure 3. Given that the image encoder contains only attention blocks for image embeddings, we insert a DConvAdapter into each transformer block in the image encoder.

Our main contributions are as follows:

- We introduce MaskSAM, a novel prompt-free SAM framework for mask classification in medical image segmentation. To our knowledge, MaskSAM is the first prompt-free SAM-based framework that retains the full structure of the original SAM.
- We design an innovative prompt generator that leverages multiple levels of feature maps from the image encoder to generate auxiliary masks and bounding boxes as prompts, thereby eliminating the need for additional prompts. It also generates auxiliary classifier tokens, which are combined with learnable global classifier tokens within the SAM mask decoder to predict semantic labels for binary masks.
- We present two specialized adapters: the 3D depth-convolution adapter (DConvAdapter) for image embeddings and the 3D depth-MLP adapter (DMLPAdapter) for prompt embeddings. These adapters are integrated into each transformer block of the image encoder and mask decoder, enabling pre-trained 2D SAM models to extract 3D information and adapt to 3D medical images.
- We conducted extensive experiments on three challenging datasets—AMOS Ji et al. (2022), ACDC Bernard et al. (2018), and Synapse Landman et al. (2015). The results demonstrate that MaskSAM achieves state-of-the-art performance, outperforming nnUNet by 2.7%, 1.7%, and 1.0% on the AMOS2022, ACDC, and Synapse datasets, respectively.

## 2 THE PROPOSED METHOD

In this section, we first review SAM. Then, we introduce the whole structure of our proposed MaskSAM. Finally, we describe each component of MaskSAM.

### 2.1 SAM PRELIMINARIES

SAM is a prompt-driven foundation model for natural image segmentation, trained on the extensive SA-1B dataset containing 1 billion masks and 11 million images. Its architecture comprises three main components: an image encoder that employs a Vision Transformer as the backbone, a prompt encoder that embeds various types of prompts (e.g., points, boxes, or text), and a lightweight mask decoder that generates masks based on image embeddings, prompt embeddings, image positional embeddings, and output tokens. For segmenting a provided 2D image, SAM requires prompts such as points or boxes, subsequently generating a single binary mask per prompt without any associated semantic labels. However, medical segmentation tasks typically involve multiple objects with distinct semantic labels within a single image, making this approach insufficient.

## 2.2 Overview of the Proposed MaskSAM

In this section and the ones that follow, we introduce the complete pipeline and individual components of MaskSAM, as illustrated in Figure 1 and Figure 4. MaskSAM retains the full structure of SAM but introduces specifically designed blocks to adapt the model from 2D natural images to 3D medical images. The architecture of MaskSAM includes a modified image encoder, a custom prompt generator, the original prompt encoder, and a modified mask decoder. Additionally, we have developed a dataset mapping process that converts multi-class labels into binary masks for each class with semantic labels, ensuring that each predicted binary mask corresponds to a single class.

## 2.3 Proposed Dataset Mapping

SAM generates a single binary mask without an associated semantic label for each prompt, whereas the typical ground truth for medical images comprises multiple classes. Each input patch image may contain several different classes, and the challenge lies in handling varying lengths of binary masks in the ground truth. Inspired by DETR Carion et al. (2020) and MaskFormer Cheng et al. (2021), our model generates a sufficient number of binary masks, with each mask dedicated to predicting a single class. We employ bipartite matching to align the predicted masks with ground truth segments accurately. As shown at the bottom of of Figure 1, our dataset mapping pipeline converts a multi-class mask into a set of binary masks with semantic labels per class.

## 2.4 Proposed Prompt Generator

To address the need for additional prompts, we introduce a prompt generator, illustrated in Figure 4(b), that automatically generates a set of auxiliary binary masks and bounding boxes instead of relying on manual prompts. We utilize both box and mask prompts, as point prompts tend to introduce instabilities detrimental to medical segmentation tasks. Leveraging the strong representation capabilities of the Vision Transformer (ViT), we extract multiple levels of feature maps from the image encoder as input to our prompt generator. The final output of the image encoder is connected to convolution layers, upsampled, and concatenated with feature maps from lower levels, resulting in feature maps with the same size as the ground truth.

A convolutional layer is then applied to adjust the channel size to a fixed number, $N$, which exceeds the maximum number of object-level binary masks in the dataset. Additionally, we extract outputs from the last convolutional layer at each level, apply adaptive average pooling to adjust the spatial dimensions to $(2, 2)$ for box queries, and concatenate all box queries. An MLP layer then adjusts the channel to $N$, resulting in $N$ learnable binary masks and bounding boxes.

There are several combinations of learnable binary masks and boxes, as illustrated in Figure 2(a)-(d). Figure 2(a) depicts a prompt generator that produces only learnable binary masks, using these masks to calculate their corresponding bounding boxes. Figure 2(b) shows a prompt generator that creates only boxes as prompts. Figure 2(c) combines both learnable binary masks as mask prompts and learnable boxes as box prompts. In Figure 2(d), the prompt generator produces learnable binary masks as mask prompts and learnable boxes, with the final box prompts obtained by averaging the bounding boxes derived from the binary masks and the learnable boxes. Based on extensive experimentation, we found that the approach shown in Figure 2(d) is the most effective, as it incorporates more information and offers greater robustness.

To address the challenge of predicting semantic labels, the prompt generator simultaneously generates a set of auxiliary classifier tokens in a similar manner to the generation of auxiliary box prompts, with the exception of using adaptive average pooling layers to adjust the spatial dimension to $(1, 1)$ for classifier tokens. These auxiliary classifier tokens are then combined with our designed learnable global classifier tokens within the mask decoder.

## 2.5 Proposed Adapters

We adopt the lightweight adapter Houlsby et al. (2019), a bottleneck architecture consisting of two fully connected (FC) layers and an activation layer in between, which we inject into each transformer block during fine-tuning. Unlike classic 2D natural images, many medical scans, such as MRI and

| Semantic labels | Prompts | Method | Spl. | R.Kd | L.Kd | GB | Eso. | Liver | Stom. | Aorta | IVC | Panc. | RAG | LAG | Duo. | Blad. | Pros. | Average |
|---|---|---|---|---|---|---|---|---|---|---|---|---|---|---|---|---|---|---|
| ✔ | – | TransBTS Wang et al. (2021) | 0.885 | 0.931 | 0.916 | 0.817 | 0.744 | 0.969 | 0.837 | 0.914 | 0.855 | 0.724 | 0.630 | 0.566 | 0.704 | 0.741 | 0.650 | 0.792 |
| | | UNETR Hatamizadeh et al. (2022) | 0.926 | 0.936 | 0.918 | 0.785 | 0.702 | 0.969 | 0.788 | 0.893 | 0.828 | 0.732 | 0.717 | 0.554 | 0.658 | 0.683 | 0.722 | 0.762 |
| | | nnFormer Zhou et al. (2021) | 0.935 | 0.904 | 0.887 | 0.836 | 0.712 | 0.964 | 0.798 | 0.901 | 0.821 | 0.734 | 0.665 | 0.587 | 0.641 | 0.744 | 0.714 | 0.790 |
| | | SwinUNETR Hatamizadeh et al. (2021) | 0.959 | 0.960 | 0.949 | **0.894** | 0.827 | 0.979 | 0.899 | 0.944 | 0.899 | 0.828 | 0.791 | 0.745 | 0.817 | 0.875 | 0.841 | 0.880 |
| | | nn-UNet Isensee et al. (2019) | **0.965** | 0.959 | 0.951 | 0.889 | 0.820 | 0.980 | 0.890 | 0.948 | 0.901 | 0.821 | 0.785 | 0.739 | 0.806 | 0.869 | 0.839 | 0.878 |
| ✗ | nnUNet | SAM Kirillov et al. (2023) 1 point | 0.001 | 0.000 | 0.051 | 0.000 | 0.002 | 0.003 | 0.010 | 0.018 | 0.019 | 0.012 | 0.000 | 0.008 | 0.007 | 0.005 | 0.017 | 0.011 |
| ✗ | nnUNet | SAM Kirillov et al. (2023) 1 bbox | 0.679 | 0.741 | 0.640 | 0.168 | 0.443 | 0.773 | 0.671 | 0.651 | 0.554 | 0.434 | 0.232 | 0.324 | 0.444 | 0.698 | 0.602 | 0.538 |
| ✗ | nnUNet | MedSAM Ma et al. (2024) 1 point | 0.000 | 0.000 | 0.078 | 0.000 | 0.008 | 0.008 | 0.014 | 0.010 | 0.025 | 0.024 | 0.009 | 0.000 | 0.012 | 0.008 | 0.023 | 0.020 |
| ✗ | nnUNet | MedSAM Ma et al. (2024) 1 bbox | 0.714 | 0.811 | 0.702 | 0.193 | 0.469 | 0.759 | 0.725 | 0.701 | 0.681 | 0.434 | 0.365 | 0.412 | 0.462 | 0.783 | 0.758 | 0.600 |
| ✔ | No needs | SAMed Zhang & Liu (2023) | 0.849 | 0.857 | 0.830 | 0.573 | 0.733 | 0.894 | 0.816 | 0.855 | 0.784 | 0.727 | 0.622 | 0.683 | 0.701 | 0.844 | 0.829 | 0.773 |
| ✔ | No needs | SAM3D Bui et al. (2024) | 0.796 | 0.863 | 0.871 | 0.428 | 0.711 | 0.908 | 0.833 | 0.878 | 0.749 | 0.699 | 0.564 | 0.607 | 0.635 | 0.884 | 0.850 | 0.752 |
| ✔ | No needs | MaskSAM (Ours) | 0.963 | **0.973** | **0.969** | 0.872 | **0.876** | **0.982** | **0.940** | **0.962** | **0.922** | **0.888** | **0.794** | **0.813** | **0.851** | **0.920** | **0.854** | **0.905** |

Table 1: The comparison of MaskSAM with SOTA methods on the AMOS testing dataset evaluated by Dice Score. To fair comparison, all results are based on 5-fold cross-validation without any ensembles. "Semantic labels" indicate the model's ability for semantic labeling, while "Prompt" specifies the source of the prompt. The best results are indicated as in **bold**.

CT, are 3D volumes with additional depth dimensions. To incorporate this extra depth information, we introduce learnable layers into the adapters to handle the depth dimension.

In SAM, both the image encoder and mask decoder contain transformer blocks where adapters can be inserted. The mask decoder includes two types of attention blocks: one for prompt embeddings and another for image embeddings. The original adapter processes only the last (channel) dimension, which limits its ability to capture relationships between tokens. Since image embeddings contain crucial spatial information, understanding spatial relationships is essential for the model. To address this, we designed a 3D depth-convolution adapter (DConvAdapter), as shown in Figure 3(a), which adds a 3D depth-wise convolution layer in the middle of the original adapter, along with a skip connection, for all attention blocks processing image embeddings in the mask decoder.

For the remaining attention blocks that handle prompt embeddings, only a learnable block in the depth dimension is needed, as prompt embeddings lack spatial relationships. Hence, we designed a 3D depth-MLP adapter (DMLPAdapter), depicted in Figure 3(b). This adapter incorporates an inverted-bottleneck architecture consisting of two FC layers and an activation layer to process the depth dimension within the original adapter, along with a skip connection, allowing it to learn additional depth information.

Since the image encoder contains only attention blocks for image embeddings, we insert the DConvAdapter into each transformer block of the image encoder. Figure 3(c) illustrates how we insert adapters into the vision transformers, placing an adapter after the multi-head attention block and in parallel with the MLP block.

## 2.6 Modified Image Encoder

Figure 4(a) illustrates the redesigned image encoder. i) SAM works on natural images that have 3 channels for RGB while medical images have varied modalities as channels. There are gaps between the varied modalities of medical images and the RGB channels of natural images. Therefore, we design a sequence of convolutional layers to an invert-bottleneck architecture to learn the adaption from the varied modalities with any size to 3 channels. ii) The image encoder includes one positional embedding. To better understand the extra depth information, we can insert a learnable depth positional embedding with the original positional embedding. iii) Since we use the base ViT backbone, it contains 12 attention blocks. We insert our designed DConvAdapter blocks into each attention block. We extract the feature maps of each three attention blocks and the final output of the image encoder for the prompt generator.

## 2.7 Modified Image Encoder

Figure 4(a) illustrates the redesigned image encoder.

i) While SAM operates on natural images with 3 RGB channels, medical images have varied modalities that differ significantly from RGB channels. To bridge this gap, we introduce a sequence of convolutional layers within an inverted-bottleneck architecture to adapt these varied modalities, regardless of their size, into a standardized 3-channel format.

ii) The original image encoder utilizes a single positional embedding. To incorporate additional depth information crucial for medical imaging, we introduce a learnable depth positional embedding alongside the original positional embedding.

| Semantic labels | Prompts | Method | Aorta | GB | L.Kd | R.Kd | Liv. | Panc. | Spl. | Stom. | DSC |
|---|---|---|---|---|---|---|---|---|---|---|---|
| ✔ | – | TransUNet Chen et al. (2021) | 87.23 | 63.16 | 81.87 | 77.02 | 94.08 | 55.86 | 85.08 | 75.62 | 77.48 |
| | | SwinUNet Cao et al. (2021) | 85.47 | 66.53 | 83.28 | 79.61 | 94.29 | 56.58 | 90.66 | 76.6 | 79.13 |
| | | UNETR Hatamizadeh et al. (2022) | 89.99 | 60.56 | 85.66 | 84.80 | 94.46 | 59.25 | 87.81 | 73.99 | 79.56 |
| | | nnUNet Isensee et al. (2019) | 92.39 | 71.71 | 86.07 | **91.46** | 95.84 | 82.92 | 90.31 | 79.01 | 86.21 |
| | | nnFormer Zhou et al. (2021) | **92.40** | 70.17 | 86.57 | 86.25 | 96.84 | **83.35** | 90.51 | 86.83 | 86.57 |
| ✗ | GT | SAM Kirillov et al. (2023) 1 point | 1.18 | 8.33 | 0.00 | 0.16 | 1.80 | 2.10 | 0.44 | 1.17 | 1.90 |
| ✗ | GT | SAM Kirillov et al. (2023) 1 bbox | 60.05 | 24.90 | 68.87 | 54.22 | 76.91 | 45.36 | 69.20 | 67.93 | 58.43 |
| ✗ | GT | MedSAM Ma et al. (2024) 1 point | 0.93 | 8.33 | 0.00 | 5.11 | 0.69 | 2.73 | 0.00 | 1.18 | 2.37 |
| ✗ | GT | MedSAM Ma et al. (2024) 1 bbox | 70.16 | 22.44 | 79.08 | 64.63 | 76.38 | 52.31 | 73.07 | 79.10 | 64.65 |
| ✔ | No needs | SAMed Zhang & Liu (2023) | 87.77 | 69.11 | 80.45 | 79.95 | 94.80 | 72.17 | 88.72 | 82.06 | 81.88 |
| ✔ | No needs | SAMed_s Zhang & Liu (2023) | 83.62 | 57.11 | 79.63 | 78.92 | 93.98 | 65.66 | 85.81 | 77.49 | 77.78 |
| ✔ | No needs | SAM3D Bui et al. (2024) | 89.57 | 49.81 | 86.31 | 85.64 | 95.42 | 69.32 | 84.29 | 76.11 | 79.56 |
| ✔ | No needs | MaskSAM (Ours) | 91.75 | **72.20** | **87.32** | 88.15 | **97.21** | 79.62 | **92.47** | **89.11** | **87.23** |

Table 2: The comparison of MaskSAM with SOTA methods on Synapse dataset. "Semantic labels" indicate the model's ability for semantic labeling, while "Prompt" specifies the source of the prompt.

iii) Our model employs the base ViT backbone, which comprises 12 attention blocks. We insert our DConvAdapter blocks into each of these attention blocks. Additionally, we extract feature maps from every third attention block, as well as from the final output of the image encoder, for use in the prompt generator.

## 2.8 Losses and Matching

Following the methods in Cheng et al. (2021); Carion et al. (2020), we construct an auxiliary loss that includes a combination of binary cross-entropy and dice loss for auxiliary binary mask predictions ($\mathcal{L}mask^{\text{aux}}$), as well as an $L_1$ loss and generalized IoU loss Rezatofighi et al. (2019) for bounding box predictions ($\mathcal{L}box$), in our prompt generator. Additionally, we create a loss comprising the standard classification cross-entropy (CE) loss for class predictions and a combination of binary cross-entropy and dice loss for final binary mask predictions ($\mathcal{L}_{mask}^{\text{final}}$) for the final output of MaskSAM.

To identify the lowest cost assignment, we apply bipartite matching Cheng et al. (2021); Carion et al. (2020) between the ground truths and the combined set of auxiliary and final predictions. Our model then selects the matching indices from $N$ auxiliary binary masks and $N$ final binary masks through bipartite matching, using these indices to obtain specific predictions for loss calculation with the ground truths. Specifically, the desired final output $z = \{(p_i, m_i)\}_{i=1}^N$ contains $N$ pairs of binary masks $\{m_i^{\text{final}} | m_i^{\text{final}} \in [0,1]^{H \times W}\}_{i=1}^N$ with classes of probability distribution $p_i \in \Delta^{K+1}$, which contains $K$ category labels with an auxiliary "no object" label ($\varnothing$). Meanwhile, our model produces $N$ pairs of auxiliary boxes and masks, $z_{\text{aux}} = \{(b_i^{\text{aux}}, m_i^{\text{aux}}) | b_i^{\text{aux}} \in [0,1]^4, m_i^{\text{aux}} \in \{0,1\}^{H \times W}\}_{i=1}^N$. Additionally, the set of $N^{gt}$ ground truth segments $z^{gt} = \{(c_i^{gt}, b_i^{gt}, m_i^{gt}) | c_i^{gt} \in \{1, ..., K\}, b_i^{gt} \in [0,1]^4, m_i^{gt} \in \{0,1\}^{H \times W}\}_{i=1}^{N^{gt}}$ is required. Since we set $N \leq N_{gt}$ and pad the set of ground truth labels with "no object token" $\varnothing$ to allow one-to-one matching. To train the model parameters, given a matching $\sigma$, the main loss $\mathcal{L}_{\text{mask-box-cls}}$ is expressed as follows,

$$\mathcal{L}_{\text{mask-box-cls}} = \sum_{j=1}^N [-\log p_\sigma(j)(c_j^{gt}) + \mathbb{1}_{c_j^{gt} \neq \varnothing} \mathcal{L}_{mask}^{\text{aux}}(m_\sigma^{\text{aux}}(j), m_j^{gt})$$
$$+ \mathbb{1}_{c_j^{gt} \neq \varnothing} \mathcal{L}_{box}(b_\sigma(j), b_j^{gt}) + \mathbb{1}_{c_j^{gt} \neq \varnothing} \mathcal{L}_{mask}^{\text{final}}(m_\sigma^{\text{final}}(j), m_j^{gt})]. \quad (1)$$

## 3 Experiments

**Datasets and Evaluation Metrics.** We used three publicly available datasets: AMOS22 Abdominal CT Organ Segmentation Ji et al. (2022), Synapse Multiorgan Segmentation Landman et al. (2015), and the Automatic Cardiac Diagnosis Challenge (ACDC) Bernard et al. (2018). **(i) AMOS22 Dataset:** The AMOS22 dataset comprises 200 abdominal CT scan cases with annotations for 16 anatomies, including the spleen, right kidney, left kidney, gallbladder, esophagus, liver, stomach, aorta, inferior vena cava, pancreas, right adrenal gland, left adrenal gland, duodenum, bladder, and prostate/uterus. We evaluated our model on the AMOS22 leaderboard using the 200 testing images. **(ii) Synapse Dataset:** This dataset contains 30 cases of abdominal CT scans. Following the split strategy in Chen et al. (2021), we randomly split the data into 18 training cases and 12 validation cases. Model performance was evaluated using the average Dice Score (DSC) across 8 abdominal

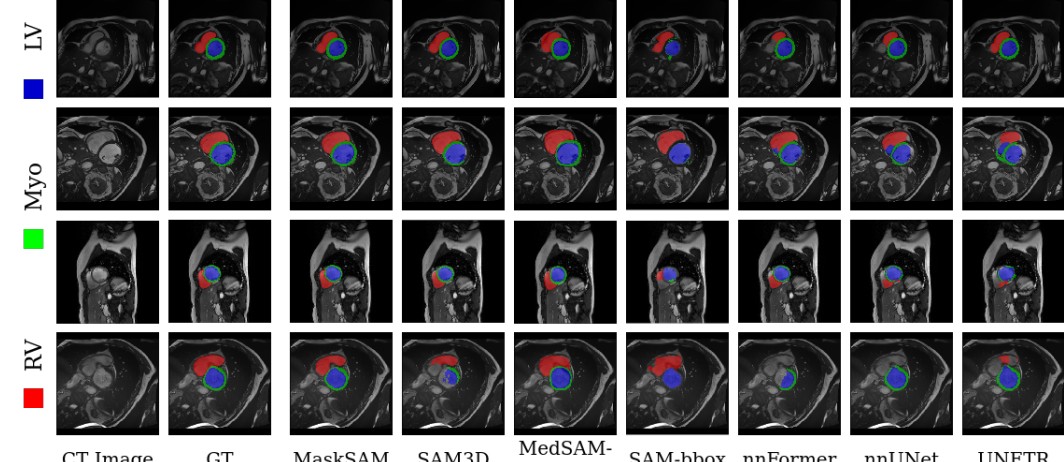

Figure 5: Qualitative comparison on the ACDC medical image segmentation dataset. MaskSAM is the most precise for each class and has fewer segmentation outliers

organs: aorta, gallbladder, spleen, left kidney, right kidney, liver, pancreas, and stomach. **(iii) ACDC Dataset:** The ACDC dataset consists of 100 patient cases, with segmentation targets including the right ventricle cavity, myocardium of the left ventricle, and left ventricle cavity. The labels cover the right ventricle (RV), myocardium (MYO), and left ventricle (LV). We randomly split the data into 70 training cases, 10 validation cases, and 20 testing cases. Model performance was assessed using the average DSC.

In Table 1, 2, 3, "Semantic labels" indicate the model's ability to perform semantic labeling, while "Prompt" specifies the source of the prompt. As SAM and MedSAM do not inherently predict semantic labels and require additional prompts, we utilized ground truth (GT) data or inferred predictions from a pre-trained nnUNet to generate prompts, using the same predictions' labels as the semantic labels.

## 3.1 COMPARISON WITH STATE-OF-THE-ART METHODS

**Results on the AMOS 2022 Dataset:** We present the quantitative results of our experiments on the AMOS 2022 dataset in Table 1, comparing our proposed MaskSAM with several widely recognized segmentation methods, including convolution-based methods (nnUNet Isensee et al. (2019)), transformer-based methods (UNETR Hatamizadeh et al. (2022), SwinUNETR Hatamizadeh et al. (2021), and nnFormer Zhou et al. (2021)), as well as SAM-based methods (SAM Kirillov et al. (2023), MedSAM Ma et al. (2024), SAMed Zhang & Liu (2023), and SAM3D Bui et al. (2024)). For a fair comparison, all methods were evaluated using 5-fold cross-validation without any ensembling. Our observations indicate that MaskSAM outperforms all existing methods on most organs, establishing a new state-of-the-art in terms of the Dice Similarity Coefficient (DSC). Additionally, we noted that using point prompts resulted in lower performance compared to using box prompts. When utilizing predictions from nnUNet for bounding box prompts, SAM and MedSAM showed decreases in accuracy by 34% and 27%, respectively, compared to nnUNet's accuracy of 87.8%, highlighting a negative impact on their results. Notably, MaskSAM outperforms nnUNet and SwinUNETR by 2.7% and 2.5% in DSC, respectively, and surpasses SAMed and SAM3D by 13% and 15% in DSC, respectively. These findings demonstrate that MaskSAM achieves state-of-the-art performance on the challenging AMOS 2022 dataset, confirming the effectiveness of our method.

**Results on the ACDC Dataset:** Table 3 presents the quantitative results of our experiments on the ACDC dataset. We compare the proposed MaskSAM with several leading methods, including SAM-based methods (e.g., SAM3D Bui et al. (2023)), convolution-based methods (e.g., R50-U-Net Ronneberger et al. (2015) and nnUNet Isensee et al. (2019)), and transformer-based meth-

| Semantic labels | Prompts | Method | RV ↑ | Myo ↑ | LV ↑ | DSC ↑ |
|---|---|---|---|---|---|---|
| | | UNETR Hatamizadeh et al. (2022) | 85.29 | 86.52 | 94.02 | 88.61 |
| | | TransNet Chen et al. (2021) | 88.86 | 84.54 | 95.73 | 89.71 |
| ✔ | – | SwinUNet Cao et al. (2021) | 88.55 | 85.62 | 95.83 | 90.00 |
| | | nnUNet Isensee et al. (2019) | 90.24 | 89.24 | 95.36 | 91.61 |
| | | nnFormer Zhou et al. (2021) | 90.94 | 89.58 | 95.65 | 92.06 |
| ✗ | GT | SAM Kirillov et al. (2023) 1 point | 3.33 | 3.20 | 27.47 | 11.33 |
| ✗ | GT | SAM Kirillov et al. (2023) 1 bbox | 70.95 | 33.33 | 81.31 | 62.53 |
| ✗ | GT | MedSAM Ma et al. (2024) 1 point | 1.88 | 4.90 | 9.48 | 5.42 |
| ✗ | GT | MedSAM Ma et al. (2024) 1 bbox | 85.86 | 80.31 | 92.33 | 86.17 |
| ✔ | No needs | SAM3D Bui et al. (2024) | 89.44 | 87.12 | 94.67 | 90.41 |
| ✔ | No needs | MaskSAM (Ours) | **92.30** | **91.37** | **96.49** | **93.39** |

Table 3: The comparison of MaskSAM with SOTA methods on ACDC dataset (DSC in %). The best are in **bold**.

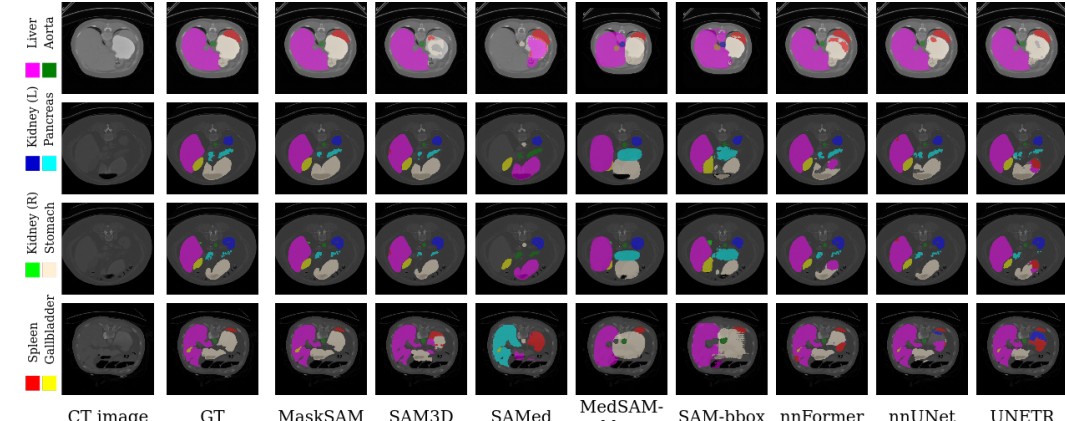

Figure 6: Qualitative comparison on the Synapse medical image segmentation dataset. MaskSAM is the most precise for each class and has fewer segmentation outliers.

ods (e.g., TransUNet Chen et al. (2021),
SwinUNet Cao et al. (2021), and LeViT-UNet-384s Xu et al. (2021)). The results indicate that MaskSAM outperforms various state-of-the-art approaches, surpassing nnFormer by 1.3%, 2.3%, 1.8%, and 0.8% in overall DSC, right ventricle (RV) dice, myocardium (Myo) dice, and left ventricle (LV) dice, respectively. Furthermore, our method outperforms SAM-based methods—SAM (1 bbox), MedSAM (1 bbox), and SAM3D—by 31%, 7%, 6%, and 3%, respectively, demonstrating the superior effectiveness of MaskSAM. In Figure Figure 5, we provide qualitative comparisons with several state-of-the-art methods, showing that MaskSAM achieves highly accurate predictions across all labels, even in the challenging and densely saturated dataset. These results validate the efficacy of our approach, as our proposed modules effectively address the limitations of SAM in adapting to medical image segmentation.

**Results on the Synapse Dataset:** Table 2 presents the quantitative results of our experiments on the Synapse dataset, comparing our proposed MaskSAM with several leading methods, including SAM-based methods (e.g., SAMed Zhang & Liu (2023) and SAM3D Bui et al. (2023)), convolution-based methods (e.g., VNet Ronneberger et al. (2015) and nnUNet Isensee et al. (2019)), and transformer-based methods (e.g., TransUNet Chen et al. (2021), SwinUNet Cao et al. (2021), and nnFormer Zhou et al. (2021)). The results indicate that MaskSAM outperforms all existing methods, achieving a new state-of-the-art performance. Specifically, MaskSAM surpasses nnFormer by 0.7% in DSC for this challenging dataset. Additionally, it outperforms the SAM-based methods—SAM (1 bbox), Med-SAM (1 bbox), SAMed, and SAM3D—by 29%, 23%, 6%, and 8%, respectively, demonstrating the superior effectiveness of our approach. Notably, our model excels in predicting large-size labels such as Liver,' Spleen,' and Stomach,' which can be attributed to our innovative DConvAdapter and DMLPAdapter. These adapters enhance the model's ability to learn intricate 3D spatial information, effectively adapting the 2D SAM framework to medical image segmentation. Figure 6 provides qualitative comparisons with representative methods, further demonstrating MaskSAM's ability to accurately predict Liver,' Spleen,' and Stomach' labels. In summary, both the quantitative and qualitative results robustly confirm the effectiveness of our method.

## 3.2 ABLATION STUDY

**Baseline Models:** The proposed MaskSAM includes 9 baseline models (denoted as B1 through B9), as summarized in Table 4. Each baseline model comprises the complete SAM structure, a prompt generator, and a learnable class token. (i) B1 utilizes a prompt generator that only produces learnable binary masks, using them as mask prompts to calculate bounding boxes as box prompts, as shown in Figure 2(a). (ii) B2 employs a prompt generator that generates only learnable boxes as prompts, as depicted in Figure 2(B). (iii) B3 uses a prompt generator that creates both learnable binary masks as mask prompts and learnable boxes as box prompts, illustrated inFigure 2(C). (iv) B4 adopts a prompt generator that generates learnable binary masks as mask prompts and learnable boxes, averaging the bounding boxes calculated from the binary masks with the learnable boxes to produce

the final box prompts, as shown in Figure 2(D). (v) B5 extends B4 by incorporating depth positional embedding blocks (DPosEmbed) into the image encoder and mask decoder. (vi) B6 further modifies B5 by enhancing the vanilla adapter with an inverted-bottleneck depth MLP and a skip connection, added after the fully connected layers for upsampling. (vii) B7 builds on B5 by inserting the inverted-bottleneck depth MLPs with a skip connection before the fully connected layers for downsampling. (viii) B8 replaces the vanilla adapter in B5 with our custom-designed DMLPAdapter. (ix) B9 represents our complete model, named MaskSAM, as illustrated in Figure 1. B9 incorporates the DMLPAdapter for prompt embeddings and the DConvAdapter for image embeddings, building upon B8.

**Ablation Analysis:** The results of the ablation study are presented in Table 4. When using our proposed MaskAvgB-BoxPG to first generate auxiliary binary masks and auxiliary boxes, and then averaging the bounding boxes derived from the auxiliary binary masks with the learnable auxiliary boxes to form the final box prompts, the model achieves the best performance. It shows improvements of 1.9%,

| | Method | DSC ↑ |
|---|---|---|
| B1 | SAM + MaskPG + vAdapter | 89.53 |
| B2 | SAM + BBoxPG + vAdapter | 88.78 |
| B3 | SAM + MaskBBoxPG + vAdapter | 90.08 |
| B4 | SAM + MaskAvgBBoxPG + vAdapter | 91.45 |
| B5 | SAM + MaskAvgBBoxPG + vAdapter + DPosEmbed | 91.61 |
| B6 | SAM + MaskAvgBBoxPG + vAdapter w/ D-MLP after FC-Up + DPosEmbed | 92.88 |
| B7 | SAM + MaskAvgBBoxPG + vAdapter w/ D-MLP before FC-Down + DPosEmbed | 92.93 |
| B8 | SAM + MaskAvgBBoxPG + DMLPAdapter + DPosEmbed | 93.10 |
| B9 | Our Full Model (B8 + DConvAdapter) | **93.39** |

Table 4: Ablation studies of proposed methods on ACDC. {}PG means a prompt generator with different prompts. vAdapter means vanilla adapter. D-MLP means MLP layers on depth dimension. DPosEmbed means depth positional embedding.

2.6%, and 1.3% compared to B1 (using a learnable mask prompt generator), B2 (using a learnable box prompt generator), and B3 (using both a learnable mask and box prompt generator), respectively. This result confirms the effectiveness of our proposed prompt generator.

Inserting depth positional embedding (DPosEmbed) into the image encoder and mask decoder in B5 results in more than a 0.15% improvement over B4, demonstrating the value of DPosEmbed blocks.

The DMLPAdapter (B8), which involves inserting depth MLPs with a skip connection in the middle of the vanilla adapter, achieves the highest performance. It shows improvements of 0.2% and 0.1% compared to B6 (where depth MLPs are inserted with a skip connection after the fully connected layers for upsampling) and B7 (where depth MLPs are inserted with a skip connection before the fully connected layers for downsampling), respectively. This confirms the effectiveness of the proposed DMLPAdapter.

B9 represents our complete model, MaskSAM, which utilizes the DMLPAdapter for prompt embeddings and the DConvAdapter, where we replace the depth MLP with a 3D depth-wise convolution layer from the DMLPAdapter for image embeddings, as shown in Figure 1. Compared to B8, our full model achieves approximately a 0.3% improvement, demonstrating the overall effectiveness of the proposed MaskSAM.

## 4 CONCLUSION

In this paper, we introduce MaskSAM, a mask classification prompt-free SAM adaptation framework for medical image segmentation that adapts pre-trained SAM models from 2D natural images to 3D medical images without requiring any prompts. We achieve this by designing a prompt generator integrated with the SAM image encoder to generate auxiliary classifier tokens, binary masks, and bounding boxes. Each pair of auxiliary mask and box prompts addresses the need for additional prompts and is linked to class label predictions through the sum of auxiliary classifier tokens and learnable global classifier tokens within SAM's mask decoder, enabling semantic label predictions. We further enhance our model by incorporating a 3D depth-convolution adapter (DConvAdapter) for image embeddings and a 3D depth-MLP adapter (DMLPAdapter) for prompt embeddings into each transformer block of the image encoder and mask decoder. This enables pre-trained 2D SAM models to effectively extract 3D information and adapt to 3D medical images. Our method achieves state-of-the-art performance, with a Dice score of 90.52% on the AMOS2022 Ji et al. (2022) dataset, representing a 2.7% improvement over nnUNet. Additionally, MaskSAM outperforms nnUNet by 1.7% on the ACDC Bernard et al. (2018) dataset and by 1.0% on the Synapse Landman et al. (2015) dataset.

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

## A  APPENDIX

### A.1  RELATED WORK

**Deep Learning Methods for Medical Image Segmentation.** Deep learning methodologies have dominated medical image segmentation recently. The predominant approaches can be categorized into convolution-based models Ronneberger et al. (2015); Zhou et al. (2018); Milletari et al. (2016); Çiçek et al. (2016); Isensee et al. (2019), transformer-based models Zhou et al. (2021), and hybrid models Chen et al. (2021); Cao et al. (2021) that integrate convolutional and transformer architectures. The adoption of encoder-decoder networks, particularly U-shaped networks, has dominated the prevailing trend. These methods stand out by explicitly tailoring architectures for medical image segmentation and training from scratch. However, the models exhibit high inductive bias, which also limits their adaptability and overall capacity.

**Foundation Models and Parameter-efficient Finetuning.** Foundation models Brown et al. (2020); OpenAI (2023a) are dedicated to the development of large-scale, general-purpose language and vision models. These models derive a wide range of downstream applications, achieving remarkable success following the pre-training and fine-tuning paradigm He et al. (2019); Hu et al. (2021). The goal of parameter-efficient finetuning Houlsby et al. (2019); Pan et al. (2022); Hu et al. (2021); Yang et al. (2023) is to decrease the number of trainable parameters and reduce the computation cost while achieving or surpassing the performance of full finetuning. Recently, the Segment Anything Model (SAM) Kirillov et al. (2023), pre-trained over 1 billion masks on 11 million natural images, was proposed as a visual foundation model for image segmentation and has gained a lot of attention. In this paper, we adopt the strategy of parameter-efficient finetuning to adapt SAM from 2D natural image segmentation to 3D medical image segmentation.

**SAM-based Medical Image Segmentation.** SAM-based medical image segmentation research can be broadly divided into two primary streams. The first stream of studies Deng et al. (2023b); Hu & Li (2023); Zhou et al. (2023); Mohapatra et al. (2023); Roy et al. (2023); Wang et al. (2023); He et al. (2023) focuses on evaluating SAM's performance across various medical image segmentation tasks and modalities using manually provided prompts. These evaluations, conducted across different datasets, reveal that SAM performs well with certain objects and modalities compared to state-of-the-art methods. However, SAM's performance is often suboptimal in scenarios with weak

boundaries, low contrast, or small and irregular shapes, limiting its effectiveness for many medical imaging tasks. The second stream of research Ma & Wang (2023); Wu et al. (2023); Li et al. (2023); Gong et al. (2023) aims to adapt SAM more effectively for medical image segmentation tasks. The challenge lies in addressing the need for prompts. While some studies attempt to bypass this requirement by modifying or removing the prompt encoder or mask decoder, such approaches risk disrupting SAM's cohesive structure and diminishing the strengths of these components.

To overcome these limitations, we propose a prompt-free SAM framework that generates auxiliary masks and bounding boxes directly through the image encoder using a specially designed prompt generator. This approach not only retains SAM's zero-shot capabilities but also eliminates the dependence on additional prompts, providing a more comprehensive and efficient solution for medical image segmentation.

### A.2 IMPLEMENTATION DETAILS

We utilize some data augmentations such as rotation, scaling, Gaussian noise, Gaussian blur, brightness, and contrast adjustment, simulation of low resolution, gamma augmentation, and mirroring. We set the initial learning rate at 0.001 and employ a "poly" decay strategy in Eq. equation 2.

$$lr(e) = init\_lr \times (1 - \frac{e}{\text{MAX\_EPOCH}})^{0.9}, \tag{2}$$

where $e$ means the number of epochs, MAX_EPOCH means the maximum of epochs, set it to 1000 and each epoch includes 250 iterations. We use SGD as our optimizer and set the momentum to 0.99. The weighted decay is set to 3e-5. All experiments are conducted using single NVIDIA RTX A6000 GPUs with 40GB memory.

**Deep Supervision.** Our network is trained with deep supervision when training. Auxiliary losses are added in the decoder to the last three stages (the three largest resolutions). For each deep supervision output, we downsample the ground truth segmentation mask for the loss computation with each deep supervision output. The final training objective is the sum of all resolutions loss:

$$\mathcal{L} = w_1 \cdot \mathcal{L}_1 + w_2 \cdot \mathcal{L}_2 + w_3 \cdot \mathcal{L}_3 \tag{3}$$

where the weights halve with each decrease in resolution (*i.e.,* $w_2 = \frac{1}{2} \cdot w_1$; $w_3 = \frac{1}{4} \cdot w_1$, etc), and all weight are normalized to sum to 1. Meanwhile, the resolution of $\mathcal{L}_1$ is equal to $2 \cdot \mathcal{L}_2$ and $4 \cdot \mathcal{L}_3$

