# OpenReview forum: "MaskSAM: Towards Auto-prompt SAM with Mask Classification for Medical Image Segmentation"
_ICLR.cc/2025/Conference — ICLR 2025 Conference Withdrawn Submission_

### Official Review · Reviewer_f2fR · 2024-10-30

**Soundness:** 2
**Presentation:** 1
**Contribution:** 2
**Rating:** 3
**Confidence:** 4

**Summary:**

The paper introduces MaskSAM, a novel adaptation framework of the "Segment Anything Model" (SAM) for medical image segmentation, overcoming SAM's limitations in this field. MaskSAM addresses the unique challenges of medical image segmentation by creating a prompt-free solution that supports 3D medical images and generates semantic labels for masks. The main contributions of MaskSAM include:

Prompt-Free Mask Classification: Unlike the original SAM, MaskSAM can predict semantic labels without additional prompts. It achieves this through a new prompt generator integrated with the SAM image encoder to produce auxiliary binary masks and bounding boxes as prompts, enhancing the adaptability to complex medical images.

3D Adaptation: MaskSAM adapts the SAM’s 2D architecture to 3D medical images, incorporating custom-designed adapters like the 3D depth-convolution adapter (DConvAdapter) for image embeddings and the 3D depth-MLP adapter (DMLPAdapter) for prompt embeddings, which extract 3D features crucial for accurate medical segmentation.

**Strengths:**

1. Prompt-Free Method for Medical Image Segmentation:
- MaskSAM introduces a prompt-free segmentation scheme, which overcomes the original SAM model's dependence on prompts, making it more efficient in medical image segmentation tasks.
- Through a specially designed Prompt Generator, MaskSAM is able to generate auxiliary binary masks and bounding boxes, solving the problem that the SAM model cannot generate semantic labels in multi-category medical images. This improvement improves the adaptability of the model, making it more suitable for complex multi-category medical images.

2. Enhanced 3D image adaptation capabilities:
- Medical images (such as CT and MRI scans) are usually 3D data. MaskSAM successfully extends the SAM model from 2D images to 3D images through innovative 3D adapters (DConvAdapter and DMLPAdapter), enhancing its applicability to medical images. These adapters significantly improve the model's ability to capture three-dimensional spatial information by adding deep convolutions and deep MLPs, making it perform well in complex 3D medical image segmentation tasks.

3. Maintain the original SAM model structure and avoid model reconstruction:
- When adapting to medical images, MaskSAM retains the structure of the original SAM model without making major changes to the model framework. This design retains the zero-shot segmentation capability of the SAM model and achieves adaptation to medical images through a lightweight adapter module. This approach makes it easier for MaskSAM to migrate to the medical field without destroying the advantages of the SAM model, and has high scalability.

**Weaknesses:**

1.High computational cost: The article mentions that MaskSAM introduces multiple adapters and generators on the basis of retaining the SAM structure, but does not elaborate on its computational resource requirements and model inference speed. In practical applications, this complex structure may bring high computational costs, especially when processing 3D medical images, which requires a lot of computing resources. Therefore, in some low-computing environments, the practical application value of MaskSAM may be affected.

2.Limited applicability and innovation of the method: Although MaskSAM proposes a promptless method suitable for medical images, the core of the method is still based on the existing SAM framework, and the innovation is mainly focused on the adjustment and adaptation of SAM. The implementation of 3D adaptation (e.g., DConvAdapter and DMLPAdapter) is innovative but relatively incremental innovation on the basis of 3D data processing and does not demonstrate higher-level generality or versatility.

3.Clarity and structure issues: The current structure and presentation of the paper is somewhat confusing, which affects the readability and understanding of the MaskSAM framework. For example, Sections 2.6 and 2.7 have the same title, which may lead to confusion about the distinction and importance of these sections. In addition, the location of various adapters (e.g., DConvAdapter and DMLPAdapter) is not clearly described in the context of the architecture, making it unclear to the reader what their specific role and integration points are in the model. Providing a clear and coherent overview with clearly defined section titles and visual illustrations of the adapter locations would greatly enhance understanding and readability.

4.Generalization and scope of experiments: While the paper introduces the hint generator as a key innovation, its impact and generalization potential are still limited due to the focus on a single dataset or a narrow set of datasets. The effectiveness of MaskSAM on medical images from different domains that it has not seen has not been fully demonstrated, which is crucial to the value of establishing a generalizable hint-free segmentation model. By extending the experiments to multiple datasets from different medical imaging domains, this paper can strongly validate the broader applicability of the hint generator and justify its inclusion as a meaningful enhancement.

**Questions:**

1. The content of the article is too confusing. (For example: Why are the titles of Sections 2.6 and 2.7 the same? There is no clear description of where the various adapters are placed)

2. You can conduct experiments on multiple fields of the same medical type to prove the domain generalization ability of MaskSAM. If you only verify it on a dataset of the same domain, then the significance of Prompt Generator is limited.

---

### Official Review · Reviewer_Qp4u · 2024-11-02

**Soundness:** 2
**Presentation:** 3
**Contribution:** 3
**Rating:** 5
**Confidence:** 4

**Summary:**

The study proposes MaskSAM, an automatic prompt-free model framework for medical image segmentation. This work aims to improve the limitations of the classic Segment Anything Model (SAM), adapting it to meet the demands of multi-class labeling and 3D information in medical imaging.

**Strengths:**

1. MaskSAM achieves multi-class medical image segmentation without the need for additional prompts by generating auxiliary classifier tokens, binary masks, and bounding boxes, marking progress in addressing SAM's need for multi-class labeling.

2. The paper introduces 3D depth-convolution and depth-MLP adapters, extending SAM's 2D architecture to 3D medical imaging, which is innovative for handling volumetric data such as CT and MRI.

3. Experimental results on several medical datasets (AMOS2022, ACDC, Synapse) demonstrate that MaskSAM outperforms existing self-supervised or supervised methods, such as nnUNet, in terms of Dice score.

**Weaknesses:**

1. It is recommended that the authors consider simplifying the model’s modules without compromising performance to reduce computational complexity and improve practical applicability.

2. Conduct more detailed ablation studies on the contributions of different modules to better demonstrate the impact of the 3D adapters and prompt-free generation framework.

3. Explore MaskSAM's performance on other types of medical data (e.g., pathology slide images or other 2D image types) to verify its generalization capability across different modalities.

4. It is suggested to include more interpretative images and visualization methods to help readers understand the model’s feature-capturing capability and classification decisions.

**Questions:**

None.

---

### Official Review · Reviewer_pPir · 2024-11-02

**Soundness:** 2
**Presentation:** 2
**Contribution:** 2
**Rating:** 5
**Confidence:** 4

**Summary:**

This paper presents a novel Segment Anything variant called “MaskSAM.” The proposed prompt generator alleviates the SAM’s dependence on manual prompts during the testing phase, and the re-designed adapters effectively leverage 3D information in medical images. Experiments on public datasets demonstrate that this framework achieves good performance.

**Strengths:**

1. The writing is well and easy to follow.
2. The idea of auto-prompting the segment anything model is interesting.
3. The proposed adapters can effectively utilize the 3D information in the image.

**Weaknesses:**

1. The claim “MaskSAM is the first prompt-free SAM-based framework that retains the full structure of the original SAM” is inaccurate. There are many related works that maintain the complete structure of SAM (including prompt encoder and mask decoder) and focus on auto prompting, such as:
* AlignSAM [*1], which leverages reinforcement learning to generate prompts.
* PerSAM [*2] and Matcher [*3], which leverages feature similarity to generate prompts.
2. As this work focuses on auto prompting, it is better to visualize the class-aware bboxes generated by the model.
3. The compared methods are a bit outdated. In Table 2, the latest non-SAM method is UNETR, which is proposed in 2022. In fact, there have been some high-impact efforts in the last two years that achieved good results on the Synapse dataset. For example, UCTNet [*4] and UNETR++ [*5] achieve the mean DSC of 89.44% and 87.22% respectively. Considering that the total number of parameters of MaskSAM should also be much larger than that of UCTNet and UNETR++, the superiority of MaskSAM is unclear.
4. Which variant of SAM is this study based on? (ViT-B, ViT-L, or ViT-H)
5. Since the authors have added many additional trainable components to SAM, it is necessary to report the trainable parameters of different SAM-based methods.
6. (Optional) SAM2 [*6], the successor to SAM, was proposed at the end of July. It is encouraged to discuss some SAM2 variants that are also designed for medical image segmentation, such as [*7, *8, *9].

[*1] AlignSAM: Aligning Segment Anything Model to Open Context via Reinforcement Learning. CVPR 2024

[*2] Personalize Segment Anything Model with One Shot. ICLR 2024

[*3] Matcher: Segment Anything with One Shot Using All-Purpose Feature Matching. ICLR 2024

[*4] UCTNet: Uncertainty-guided CNN-Transformer hybrid networks for medical image segmentation. Pattern Recognition 2024

[*5] UNETR++: Delving Into Efficient and Accurate 3D Medical Image Segmentation. IEEE Transactions on Medical Imaging 2024

[*6] Sam 2: Segment Anything in Images and Videos. arXiv

[*7] Medical sam 2: Segment medical images as video via segment anything model 2. arXiv

[*8] Biomedical SAM 2: Segment Anything in Biomedical Images and Videos. arXiv

[*9] SAM2-UNet: Segment Anything 2 Makes Strong Encoder for Natural and Medical Image Segmentation. arXiv

**Questions:**

See weakness.

**Details Of Ethics Concerns:**

None.

---

### Official Review · Reviewer_rzag · 2024-11-03

**Soundness:** 2
**Presentation:** 3
**Contribution:** 2
**Rating:** 5
**Confidence:** 4

**Summary:**

This work introduces MaskSAM, a prompt-free adaptation of SAM, incorporating a learnable classifier token and a 3D depth-MLP adapter. This works aims to address the meaningful challenge of adapting the foundation model of SAM. However, the adapter design is fairly common in prior works, and the contribution seems to be limited, with only marginal improvements in both experimental results and methodology.

**Strengths:**

1. This work is well written and easy to understand.
2. This work aims to solve a meaningful problem to adapt SAM, by introducing learnable classification tokenizer and extend to 3D embedding.
3. The proposed method is simple and effective to some extent.

**Weaknesses:**

1. My primary concern is the lack of novelty. As outlined in the paper, numerous studies have already explored adaptations of this approach for medical image segmentation. This work does not present a clearly innovative idea or substantial differentiation from existing benchmarks, making its contribution appear more incremental than original.

2. The improvement in experimental performance also seems minimal. As shown in Table 1 and Table 2, the proposed method does not demonstrate significantly better results compared to existing approaches.

**Questions:**

1. Please include a figure and description that clearly illustrate the high-level novelty of this work.

2. For Figure 3, consider adding caption details to make the comparison more understandable.

3. In Tables 1 and 2, the improvement appears marginal. Please provide a description to explain this result.

---

### Note · Authors · 2024-11-15

I have read and agree with the venue's withdrawal policy on behalf of myself and my co-authors.